# Effects of COVID-19 vaccination and previous infection on Omicron SARS-CoV-2 infection and relation with serology

Brechje de Gier[1], Anne J. Huiberts [1], Christina E. Hoeve[1], Gerco den Hartog [1,2,3], Henri van Werkhoven[1,4], Rob van Binnendijk[1], Susan J. M. Hahné[1], Hester E. de Melker[1], Susan van den Hof [1] & Mirjam J. Knol [1]✉

An increasing proportion of the population has acquired immunity through COVID-19 vaccination and previous SARS-CoV-2 infection, i.e., hybrid immunity, possibly affecting the risk of new infection. We aim to estimate the protective effect of previous infections and vaccinations on SARS-CoV-2 Omicron infection, using data from 43,257 adult participants in a prospective community-based cohort study in the Netherlands, collected between 10 January 2022 and 1 September 2022. Our results show that, for participants with 2, 3 or 4 prior immunizing events (vaccination or previous infection), hybrid immunity is more protective against infection with SARS-CoV-2 Omicron than vaccine-induced immunity, up to at least 30 weeks after the last immunizing event. Differences in risk of infection are partly explained by differences in anti-Spike RBD (S) antibody concentration, which is associated with risk of infection in a dose-response manner. Among participants with hybrid immunity, with one previous pre-Omicron infection, we do not observe a relevant difference in risk of Omicron infection by sequence of vaccination(s) and infection. Additional immunizing events increase the protection against infection, but not above the level of the first weeks after the previous event.

Since the emergence of the SARS-CoV-2 Omicron variant-of-concern and its subvariants, high transmissibility and immune evasiveness have resulted in a massive global incidence of infections. In the Netherlands, the first Omicron infection was detected by sequencing in November 2021, and by January 10, 2022 Omicron BA.1 comprised 90% of sequenced infections[1]. Since that time, waves of infections have occurred in the Netherlands with the emergence of Omicron subvariants BA.1, BA.2, and BA.5. Seroprevalence data have shown a large increase up to around 60% in anti-nucleoprotein (N) seropositivity between January and May 2022, which is illustrative for the large waves of SARS-CoV-2 Omicron infections in the Netherlands[2]. Combined with the high vaccination coverage among adults in the Netherlands

(83% for primary vaccination, 64% at least one booster as of August 28, 2022[3]), the majority of the Dutch adult population has acquired 'hybrid immunity'; i.e., at least one vaccine dose plus at least one previous SARS-CoV-2 infection.

When assessing the expected benefit of additional vaccination rounds, it has therefore become more relevant to gather insight into the effect of additional immunizing events on the population with hybrid immunity, aside from vaccine effectiveness per se (i.e., the difference in risk between vaccinated and unvaccinated people). A systematic review of mainly test-negative case-control studies has found better protection of hybrid immunity against Omicron infection compared to vaccine-induced immunity, with limited data on the

[1]Center for Infectious Disease Control, National Institute for Public Health and the Environment, Bilthoven, the Netherlands. [2]Laboratory of Medical Immunology, Radboud Institute for Molecular Life Sciences, Radboudumc, Nijmegen, The Netherlands. [3]Radboud Center for Infectious Diseases, Radboudumc, Nijmegen, The Netherlands. [4]Julius Center for Health Sciences and Primary Care, University Medical Center Utrecht, Utrecht University, Utrecht, the Netherlands. ✉e-mail: Mirjam.knol@rivm.nl

added benefit of further vaccine doses[4]. Registry-based studies are becoming less suitable to study hybrid immunity, as laboratory testing has become less available and self-administered antigen testing has become more commonplace.

VASCO (VAccine Study COvid) is a 5-year prospective cohort study among approximately 45,000 community-dwelling participants in the Netherlands, which aims to estimate effectiveness of COVID-19 vaccination on SARS-CoV-2 infection[5]. By including data on circulating Nucleoprotein (N) antibodies and self-administered antigen test results, underascertainment of previous and current infections are reduced as much as possible. We analyzed the risk of Omicron infection by type of prior immunizing events (vaccine-induced, infection-induced, or hybrid immunity), and by sequence and number of prior immunizing events (vaccine doses and previous infections) within the VASCO cohort. We further explore the associations between types of prior immunizing events, circulating spike receptor binding domain (S) antibody concentrations and risk of infection.

## Results

### Study population

During the study period, from 10 January 2022 to 1 September 2022, 43,257 participants contributed 8,291,966 person-days, in which 20,418 SARS-CoV-2 infections occurred. Of these infections, 2198 (10.8%) were detected only by N-antibodies while the remaining were reported based on a positive (self) test. Of the positive tests where the type of test was reported by participants, 55% pertained PCR. As during April 2022 formal testing facilities were scaled down, 87% of reported positive tests from May 2022 onward pertained antigen tests. Of 9,727 infections occurring before the study period, 1251 (12.9%) were detected by N-antibodies only while the remaining were reported based on a positive (self) test. Characteristics of the participants are shown in Table 1. Participants between 60 and 69 years old on 1 January 2022 were overrepresented in the cohort, due to the study design with oversampling of older people. The majority of participants attained a high education level (57.1%) and most participants were female (62.9%). Figure S1 shows the development of immunity status of participants during the study period, at time points 10 January, 1 March, 1 June and 1 September 2022 in terms of type of immunity and number of prior immunizing events. Participant drop-out rate was 7% during the study period. Table S1 shows the vaccines received in the study population, per dose. Vaxzevria was received as the primary series in 30–35% of participants. For further doses, the vast majority pertained mRNA vaccines.

### Types of immunity (vaccine-induced, infection-induced, or hybrid)

Hybrid immunity consistently conferred better protection against Omicron infection than vaccine-induced immunity up to 30 weeks after the last event in all three strata (2, 3, or 4 prior immunizing events) (Fig. 1), with a reduction in hazard rate of 71–85% in the 4–10 weeks after the last event compared to only vaccine-induced immunity (aHR (95% CI): 0.16 (0.09–0.28), 0.29 (0.26–0.33), 0.15 (0.12–0.18) for 2, 3, and 4 exposures, respectively). The comparison with infection-induced immunity could only be made for participants with two immunizing events in their history, as having more than two previous infections and no vaccination was very uncommon during the study period. Within the two prior immunizing events stratum, infection-induced immunity seemed to confer higher protection compared to hybrid immunity (aHR (95% CI) for 4–10 weeks after last exposure: 0.16 (0.09–0.28) for hybrid immunity and 0.06 (0.02–0.17) for infection-induced immunity compared with vaccine-induced immunity (Data S1)), however, this difference was not statistically significant. Protection against infection decreased notably faster with time since the last event for hybrid compared to vaccine-induced immunity (Fig. S2A). For example, in the stratum with 3 prior immunizing events, hybrid immunity against infection was 80% lower after 30–40 weeks compared to 4–10 after the last event (aHR 0.20, 95% CI 0.15–0.26). For vaccine-induced immunity, this decrease was 33% (aHR 0.67, 95% CI 0.56–0.79).

The difference between types of immunity were partly explained by S-antibody concentrations. While hybrid immunity consistently protected more against infection, only approaching the level of vaccine-induced immunity (week 4–10) in the 30–40 week interval, hybrid immunity GMCs were lower after 20–30 weeks than those of initial vaccine-induced immunity (week 4–10) (Data S1). This is further shown in Fig. 1, juxtaposing aHR with 1/GMC ratio for comparability. Similar rates of S-antibody waning were seen for hybrid and vaccination-only immunity (Fig. S2B). In the stratum with 3 prior immunizing events, the decrease in S-antibody concentration for weeks 30–40 compared to weeks 4–10 was 64% (GMC ratio of 0.36) for both hybrid and vaccine-induced immunity. Of note, GMCs were much lower with infection-induced immunity compared to vaccine-induced or hybrid immunity, despite high effectiveness against infection (Data S1). These estimates are, however, based on small numbers and are therefore uncertain.

Irrespective of type or number of immunizing events, S-antibody concentration was associated with risk of infection in the 3 weeks after receipt of the serum sample in a dose-response manner, showing a 71% lower incidence of infection for persons in the highest quartile (>32,401 BAU/ml) of S-antibody concentration compared to the lowest quartile (<6778 BAU/ml; aHR (95% CI): 0.29 (0.23–0.37) (Table S2).

### Sequence of immunizing events

Among participants with hybrid immunity consisting of one prior infection and two or three vaccinations (i.e., three or four immunizing events), and excluding persons with prior Omicron infections, we compared risk of infection by the type of the first and last immunizing event. While for participants with 3 prior immunizing events, in the first 4–10 weeks after the last event, the incidence of infection was lower for vaccination first and infection last, compared to infection first and vaccination last [4,10] (aHR 0.50 (95% CI: 0.30–0.81)), for other intervals since last event and for participants with 4 immunizing events no clear advantage of either sequence of events was seen and confidence intervals were wide (Fig. 2).

## Table 1 | Characteristics of the study population

|  | n (%) |
|---|---|
| Total | 43,257 |
| Age on January 1, 2022 | |
| 18–45 | 8799 (20.3) |
| 45–59 | 11,399 (26.4) |
| 60–69 | 18,178 (42.0) |
| 70–85 | 4881 (11.3) |
| Sex | |
| Female | 27,230 (62.9) |
| Male | 16,007 (37.0) |
| Other | 20 (0.0) |
| Education level | |
| High | 24,683 (57.1) |
| Intermediate | 12,389 (28.6) |
| Low | 5945 (13.7) |
| Other | 240 (0.6) |
| Medical condition at start of the study period | |
| No | 30,195 (69.8) |
| Yes | 13,062 (30.2) |

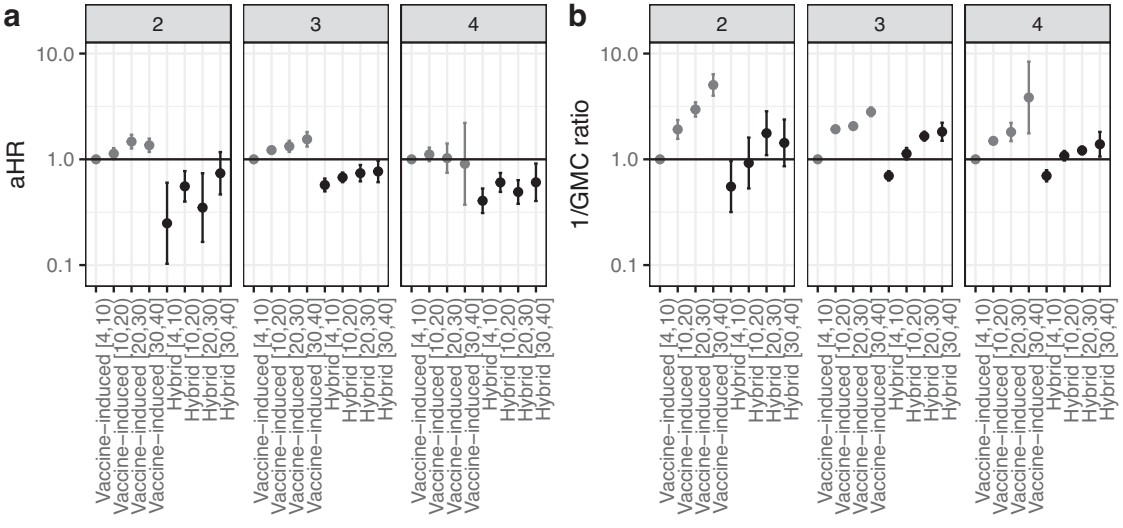

**Fig. 1 | Hazard ratio of infection and 1/ geometric mean concentrations of S-antibodies, by type of immunity and weeks since the last immunizing event.**
**a** Adjusted hazard ratio (aHR) of infection with Omicron SARS-CoV-2 by hybrid or vaccine-induced immunity, stratified by the number of prior immunizing events, $n = 39,810$ participants. **b** One divided by the adjusted geometric mean concentration (GMC) ratio of S-antibodies, $n = 20,670$ participants. In both analyses, 4 to 10 weeks after the last vaccination for vaccine-only immunity was the reference group. The group with 4 immunizing events only includes participants aged 60 and older, because younger people were not eligible for 4 vaccinations. Data are presented as aHR with 95% confidence intervals, adjusted for age, sex, educational level and medical risk group. Numbers between brackets on the x axis represent weeks since the last immunizing event. Underlying data and crude estimates can be found in Data S1.

## Additional immunizing events

We analyzed whether risk of infection further decreases after additional immunizing events. We included person-time with two to five prior immunizing events, dependent on the type of immunity and age group. Prior Omicron infections were included in this analysis. Vaccine-induced immunity was stratified by age below or above 60 years, as participants under the age of 60 were not eligible for 4 vaccinations. Figure 3 shows that among participants with hybrid immunity, in the first 4 to 10 weeks after the fourth or fifth immunizing event, no additional protection against infection was observed compared to week 4–10 after the third event, which did seem to provide additional benefit over a second immunizing event although the CI includes 1. For participants with vaccine-induced immunity, while a recent third dose showed a small benefit compared to a recent second dose among both 18–59 and >=60-year-olds, a fourth vaccine dose did not show a clear additional benefit among participants aged 60 years and older. Taken together, Fig. 3 indicates that while additional immunizing events increase protection against infection by reducing the time since the last event, a higher number of immunizing events is not in itself associated with a lower risk of infection.

## Discussion

With data from a large prospective cohort study, including data on self-administered antigen tests and serology, we estimated the protection from different types, sequences and numbers of immunizing events against SARS-CoV-2 Omicron infection. We found that, given an equal number of prior immunizing events, persons with hybrid immunity had a lower risk of infection compared to persons with vaccine-induced immunity (71–85% lower risk in weeks 4–10 post-last event, depending on the number of prior immunizing events). Our findings are in line with a recent systematic review and meta-analysis, which also found a lower risk of Omicron infection with a prior infection plus a primary series compared to booster vaccination without prior infection (60.4% versus 24.5% effectiveness, respectively)[4].

The lower risk of infection among participants with hybrid immunity was partly explained by higher mean S-antibody concentrations. This finding is in line with Wratil et al., who showed higher concentrations of Omicron-neutralizing S-antibodies in individuals with hybrid immunity compared to vaccine-induced immunity[6]. However, rapid waning of both protection against infection and of S-antibody concentrations was apparent in our study, with hybrid immunity showing faster waning of protection against infection than vaccine-induced immunity, although the higher protection was maintained up to 40 weeks post last event in most instances. With each quartile increase in S-antibody concentration, risk of infection decreased significantly. Previous studies performed preceding the Omicron variant found a reduced risk of infection in S-seropositive compared to seronegative individuals[7–9]. The dose-response relationship we observed between S-antibody concentration and risk of infection confirms the role of S-antibodies in protection against Omicron. A systematic review found humoral correlates of protection against pre-Omicron variants, but also stressed that antibody levels are not likely to correlate to full protection against infection[10]. Indeed, as the lower risk of infection with hybrid immunity was only partly reflected by S-antibody concentration, other factors likely also contribute to the protective effect of hybrid immunity. Virus neutralization capacity is not only determined by antibody levels but also by binding strength, in addition to other factors such as cellular immunity. Moreover, it is plausible that mucosal antibody concentration is more strongly correlated with protection against infection than serum antibody concentrations. A recent paper by Cohen et al. showed that mucosal concentrations of anti-Spike antibodies may decline faster than plasma concentrations[11]. If the strong protection conferred by hybrid immunity is partly attributable to mucosal antibodies, the decrease we observed in the additional protection for persons with hybrid immunity compared to vaccine-induced immunity might be due to this decline in mucosal immunity.

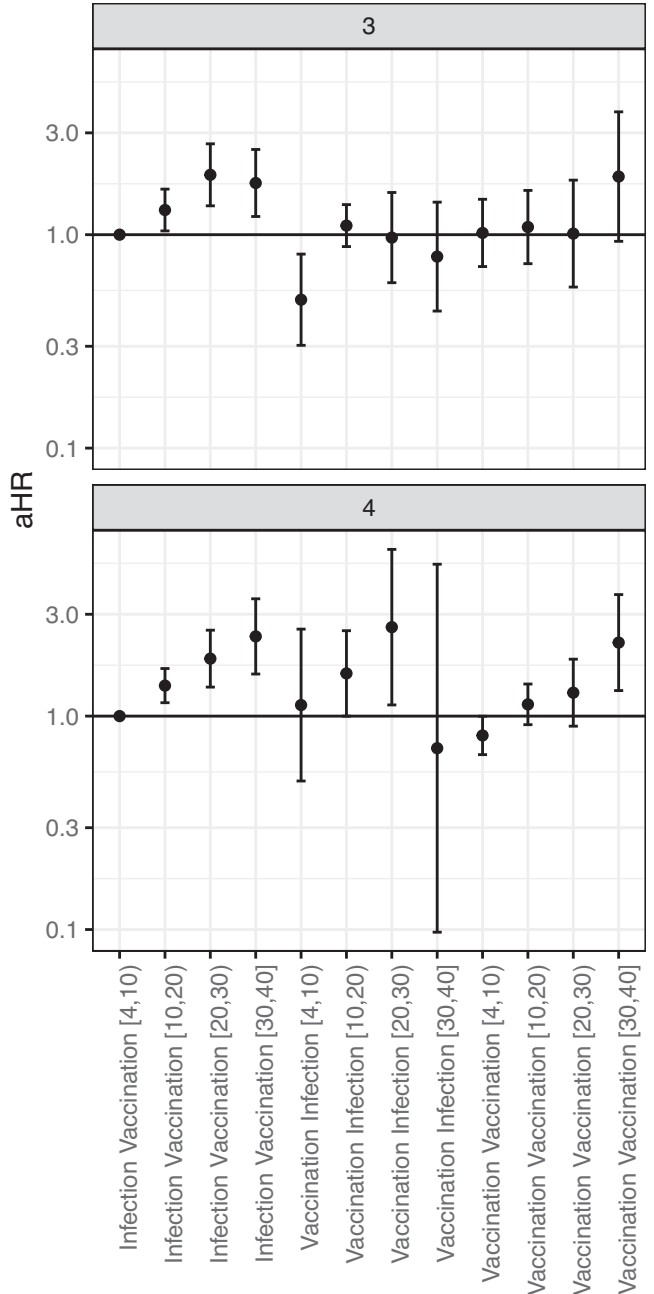

**Fig. 2 | Hazard ratio of infection by the type of first and last immunizing event and weeks since the last immunizing event.** Adjusted hazard ratio (aHR) of infection with Omicron SARS-CoV-2 by the type of first and last immunizing event, stratified by the number of exposures, *n* = 7293 participants. 4 to 10 weeks after the last vaccination for infection first and vaccination last was the reference. Data are presented as aHR with 95% confidence intervals, adjusted for age, sex, educational level and medical risk group. Numbers between brackets on the x axis represent weeks since the last immunizing event. Underlying numbers and crude estimates can be found in Data S2.

Some concerns have been voiced that the mass COVID-19 vaccination campaigns might have resulted in narrow immunity, referred to as 'original antigenic sin' or immunological imprinting[12]. We did not find evidence of a higher risk of Omicron infection in participants with hybrid immunity with a vaccination as first immunizing event compared to infection as first immunizing event. In line with this, immunological studies have thus far not found narrower responses after vaccination compared to infection[13]. We did not observe an effect of sequence of immunizing events, which indicates that the potential of

hybrid immunity is not dependent on a recent infection, and that previous infections together with vaccinations establish solid immunity, also against immune escape variants such as Omicron. This is further corroborated by the sensitivity analysis showing higher protection by hybrid immunity, also when excluding participants with a previous Omicron infection (Fig. S3). Likewise, Carazo et al. found no difference in effectiveness of hybrid immunity against Omicron infection between persons with prior infection before, in between or after vaccinations, given an equal number of doses[14].

Our results further show that additional immunizing events did not alter protection, apart from decreasing the time since the last immunizing event. This indicates that among populations with high levels of immunity, additional vaccine doses have the potential to reduce risk of any infection only temporarily. The meta-analysis by Bobrovitz et al. did find a benefit of additional vaccine doses among people with hybrid immunity, with a pooled relative protection of 40.8% (95% CI: 22.0–62.7%) after prior infection plus a first booster compared to prior infection plus a primary series[4]. Possibly, the sequential emergence of increasingly immune-evasive subvariants have counteracted the benefit of additional doses in our study period. The discrepancy might also be related to the inclusion of infections based on serology in our study, likely representing the lowest severity of SARS-CoV-2 infections. It is well established that the COVID-19 vaccines provide much higher protection against severe COVID-19 than against infection[15], and Bobrovitz et al. found also higher and more durable effectiveness of hybrid immunity against severe Omicron COVID-19 compared to vaccine-induced immunity[4].

A strength of our cohort study is the use of serology to ascertain unreported infections, and the data on self-administered antigen test results, in the analysis of hybrid immunity against SARS-CoV-2 Omicron infection. As formal community-based testing facilities have been scaled down since April 2022, 87% of reported tests included in our study from May 2022 onward pertained antigen tests. The notification of positive tests or vaccination at any time using the study app will have limited recall bias. Moreover, the combination of infection data with serology allowed comparison between patterns of risk of infection and S-antibody concentrations. Still, misclassification of both infections and vaccinations due to the self-reported nature of these data is possible and, therefore, a limitation of our study. Also, the sensitivity for detecting unreported infections by serology will have been less than 100%. Limitations of this study include the pooling of the Omicron subvariants. During the study period, sequentially Omicron BA.1, BA.2, BA.5, and to a lesser extent BA.4 caused the majority of the infections in the Netherlands. In previous Dutch studies, vaccine effectiveness was not significantly different between Omicron subvariants but previous BA.1 infections conferred lower protection to BA.4/5 compared to BA.2[16, 17]. For infections before the study period, the variant of infection is not known for certain, and variants are highly correlated with calendar time, thus complicating disentangling effects of variants and time since infection. In our analysis of the sequence of exposures, we, therefore, excluded participants with a prior infection after 20 December 2021, because from that moment on a relevant proportion of sequenced isolates pertained Omicron. A further limitation is the pooled analysis of all types of COVID-19 vaccines, while different platforms are known to result in different antibody concentrations[18]. Also, the overrepresentation of persons of older age, female sex and high education level in the VASCO cohort might reduce the generalizability of our findings to the general Dutch population[5].

The use of serology to detect untested infections also comes with limitations. Seroconversion on N is only around 85% sensitive for symptomatic SARS-CoV-2 infections in the first 2 months. This sensitivity decreases over time since infection[19]. Therefore, residual underascertainment of infections may have occurred. We imputed dates of untested infections (-12%), which might have resulted in misclassification of time since most recent exposure for some

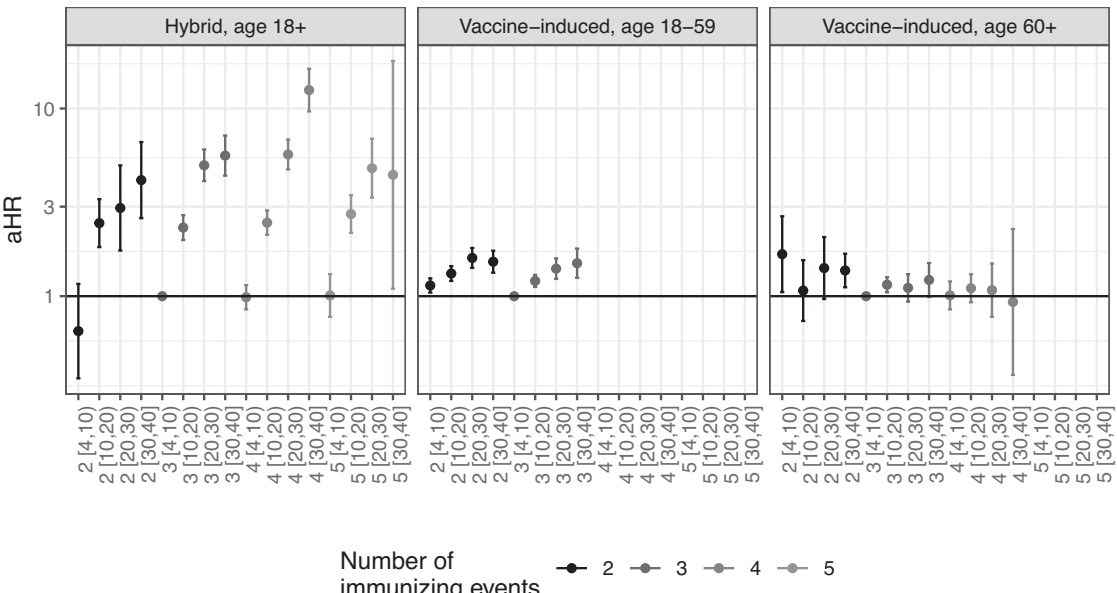

**Fig. 3 | Hazard ratio of infection by number of immunizing events and weeks since the last immunizing event.** Adjusted hazard ratio (HR) of infection with Omicron SARS-CoV-2 by number of immunizing events, $n = 42,144$ participants. 4 to 10 weeks after the third immunizing event was the reference group. Data are presented as aHR with 95% confidence intervals, adjusted for age, sex, educational level and medical risk group. Numbers between brackets on the $x$ axis represent weeks since the last immunizing event. Underlying numbers and crude estimates can be found in Data S3.

individuals and of the time of infections during the study period. We explored the potential effect of this by excluding person-time where the date of the most recent exposure was imputed and by excluding infections as outcome when the date was imputed. Neither sensitivity analyses revealed relevant deviations from our primary estimates.

In conclusion, our study shows that hybrid immunity confers better protection against SARS-CoV-2 Omicron infection than vaccine-induced immunity. This effect does not seem dependent on the sequence or number of immunizing events. It should be kept in mind that experiencing a SARS-CoV-2 infection carries significant risks, including severe COVID-19, post-covid syndrome and transmission to vulnerable people.

## Methods

### Study design and population
VASCO participants were recruited through a random draw from the population register and through (social) media campaigns and inclusion ran from May 2021 and December 2021[5]. Community-dwelling adults aged 18–85 years were included, and persons 60–85 years were oversampled. Participants completed a questionnaire at baseline including questions on demographics, COVID-19 vaccination, previous SARS-CoV-2 infections, and comorbidity. Monthly follow-up questionnaires include questions on COVID-19 vaccination, COVID-19-like symptoms and related testing behavior. After the first year of inclusion, the follow-up questionnaires are collected every 3 months instead of monthly. Participants are also asked to notify every COVID-19 vaccination and positive SARS-CoV-2 test in an online (mobile) application any time, so that near real-time data is available.

At baseline and every 6 months of follow-up participants are asked to self-collect a fingerprick sample for serology. Participants were also asked to take a fingerprick sample 1 month after primary COVID-19 vaccination series, if applicable. Since April 2022, PCR testing at community testing facilities was only advised for vulnerable groups and health care workers. Therefore, participants were provided with self-administered antigen tests to be used when having COVID-19-like symptoms or having been in close contact with someone with a SARS-CoV-2 infection.

The study period for the current analysis started on 10 January 2022, when 90% of the sequenced SARS-CoV-2 samples pertained the Omicron variant according to national pathogen surveillance[1]. The analysis was performed including information up to and including 1 September 2022.

### Ethics
The VASCO study was approved by the Medical Ethics Committee of the Stichting Beoordeling Ethiek Biomedisch Onderzoek (BEBO), Assen, the Netherlands (NL76815.056.21) and all participants gave informed consent.

### Serology
Serum samples were analyzed with the Elecsys Anti-SARS-CoV-2 S and Anti-SARS-CoV-2 electrochemiluminescence assays on the Cobas e801 (Roche Diagnostics, Mannheim, Germany), which measure immunoglobulin (Ig) levels against respectively the receptor binding domain of the Spike protein (S-antibodies) and the Nucleoprotein (N-antibodies) of SARS-COV-2. S-antibody concentrations are reported in binding antibody units (BAU/ml). Samples with higher S-antibody concentrations than the measuring range (up to 12,500 BAU/ml) were diluted 1:900, re-measured and quantified up to 225,000 BAU/ml. For N-antibody concentrations, the qualitative cut-off index (COI) was converted to numeric results in BAU/ml using batch-specific, linear calibration-lines obtained with a dilution range of the NIBSC 20/136 WHO standard (NIBSC). The cut-off for N-seropositivity was set by converting COI 1.0 to corresponding BAU/ml using these calibration lines.

### Immunizing events
COVID-19 vaccinations as well as previous SARS-CoV-2 infection (referred to as immunizing events) were included as time varying exposures. Vaccination data were self-reported, and verified with the National COVID-19 vaccination registry (CIMS) in March 2022. Participants were excluded if they received more doses than possible according to the Dutch vaccination strategy[2].

Previous SARS-CoV-2 infection was based on self-reported positive SARS-CoV-2 test, which could be a PCR test or (self-administered)

antigen test. In addition, the presence of N-antibodies was used to detect SARS-CoV-2 infection. When the first serum sample of a participant was positive for N-antibodies but no previous infection was reported, an infection date was imputed as the mid-date between the baseline questionnaire and sample receipt. When a participant had any two blood samples where the first was negative and the second was positive for N-antibodies, or where the second had at least a four-fold increase in N-antibody concentration compared to the first, and no corresponding infection was reported, an infection date was imputed as the mid-date between the two blood samples. Among all VASCO participants reporting a positive SARS-CoV-2 test since the cohort started in July 2021 and with a sample negative for N-antibodies prior to infection, 87% tested positive for N-antibodies in the first sample after infection, consistent with a previous Dutch study[19]. Among all VASCO participants reporting a positive SARS-CoV-2 test and with a sample positive for N-antibodies prior to infection, 65% had a fourfold increase in N-antibodies in the first sample after infection.

When a participant reported in the baseline questionnaire to have had a SARS-CoV-2 infection before study inclusion but without confirmation by PCR or antigen test, the infection was only included in the analysis if the first serum sample was positive for N-antibodies.

### Outcome
The outcome was any SARS-CoV-2 infection during the study period. SARS-CoV-2 infection was based on any self-reported positive SARS-CoV-2 test and presence of, or fourfold increase in, N-antibodies as described above.

### Covariates
Sex and educational level were included as fixed covariates and age and medical risk group as time-varying covariates. Educational level was defined as low (no education or primary education only), intermediate (secondary school or vocational training) and high (bachelor's or master's degree). Medical risk group (yes/no) was defined as self-reported presence of any of the following physician-diagnosed conditions: diabetes, lung disease or asthma, missing spleen, cardiovascular disease, immune deficiency, cancer, liver disease, neurological disease, renal disease, and organ or bone marrow transplantation. All analyses were adjusted for age (as natural cubic spline with 3 degrees of freedom), sex (female, male, other), educational level (low/intermediate/high) and medical risk group (yes/no).

### Statistical analysis
Person-time was stratified according to the following time-varying variables: number of any COVID-19 vaccine doses received, prior SARS-CoV-2 infection status, medical risk group status, and by weeks post-immunizing event. The first 28 days after an immunizing event (COVID-19 vaccination or SARS-CoV-2 infection) were excluded from the person-time at risk. Follow-up ended after the last questionnaire answer date plus the median interval between questionnaire answer dates, per participant, or on September 1, 2022, whichever came first.

Cox proportional hazards models with calendar time as the underlying time scale were used to estimate the association between the different prior immunizing events and the risk of SARS-CoV-2 Omicron infection, resulting in hazard ratios with 95% confidence intervals (CIs). First, we estimated the effect of type of prior immunizing events (vaccine-induced, infection-induced or hybrid immunity) by time since last event (4 up to (not including) 10, 10–20, 20–30, or 30–40 weeks) with 4–10 weeks after vaccine-induced immunity as reference category, stratified by the number of prior immunizing events (2, 3, or 4). In the stratum with four prior immunizing events only people aged 60 and older were included, as only this age group was eligible for four vaccinations in the study period.

Among participants with hybrid immunity we estimated the effect of type of the first and last prior immunizing event (vaccination or

infection) by time since last event with 4–10 weeks after vaccine-induced immunity as the reference category, stratified by the number of exposures (3 or 4). In this analysis, we excluded participants with a previous infection after 20 December 2021, as from this date Omicron and Delta co-circulated, to avoid confounding the analysis on the effect of the sequence of prior immunizing events with the protective effect of a previous Omicron infection.

Lastly, we estimated the effect of additional prior immunizing events by time since last event with 4–10 weeks after the third event as reference category, stratified by type of immunity (hybrid versus vaccine-induced) and age (18–59 and 60+) for the vaccine-induced immunity group, because only participants aged 60 and older were eligible for four vaccinations. In this analysis, prior Omicron infections were included.

We assessed whether the differences in risk of infection between vaccine-induced, infection-induced or hybrid immunity corresponded to the level of S-antibodies at similar intervals after vaccination or infection. Ratios of geometric mean concentrations (GMC) and 95% CIs were calculated using linear regression with log-transformed antibody concentrations as outcome and reference categories equal to the Cox regression models. Further, we estimated the association between S-antibody concentration (in quartiles) and risk of infection in the week of receipt of the serum sample plus the two subsequent weeks with Cox proportional hazards models with calendar time as the underlying time scale. Analyses were done with R version 4.3.0[20], using packages Epi (version 2.47.1), survival (version 3.5-5), and stats (version 4.3.0).

### Reporting summary
Further information on research design is available in the Nature Portfolio Reporting Summary linked to this article.

## Data availability
The data underlying Figs. 1–3 generated in this study have been deposited in the Data S1, S2, and S3 files. The anonymized data reported in this study can be obtained from the corresponding author upon request. The dataset may include individual data and a data dictionary will be provided. Data requests should include a proposal for the planned analyses. Decisions will be made according to data used by the statistical disclosure working group within RIVM. Data transfer will require a signed data sharing agreement.

## Code availability
Analysis code is available upon request from the corresponding author, with a response timeframe of 3 weeks.

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

## Acknowledgements
This work was supported by the Dutch Ministry of Health, Welfare, and Sport.

## Author contributions
Study design: M.J.K., S.vd.H., B.d.G., A.J.H. B.d.G. and A.J.H. performed the analyses. Data collection and study management: M.J.K., C.E.H., A.J.H., S.vd.H., H.E.d.M. Interpretation of results: B.d.G., A.J.H., C.E.H., G.d.H., H.v.W., R.v.B., S.J.M.H., H.E.d.M., S.v.d.H., M.J.K. B.d.G. and M.J.K. wrote the manuscript draft, which was reviewed and revised by all authors.

## Competing interests
The authors declare no competing interests.
