## [Peer Review File · Nature Communications]

Effects of COVID-19 vaccination and previous infection on
Omicron SARS-CoV-2 infection and relation with serologyEditorial Note: This manuscript has been previously reviewed at another journal that is not operating a transparent peer review scheme. This document only contains reviewer comments and rebuttal letters for versions considered at *Nature Communications*.

REVIEWERS' COMMENTS

Reviewer #1 (Remarks to the Author):

Thank you for the opportunity to review a revised version of this manuscript. The authors have responded well to all of my comments and made sufficient revisions of the manuscript.

Reviewer #2 (Remarks to the Author):

Dear editor,

The authors have responded to my previous comments in a satisfactory manner. There are a few small bits and pieces which would still need to be addressed better.

Abstract:

- "Among participants with hybrid immunity, with one previous pre-Omicron infection, there was no relevant difference in risk of Omicron infection by sequence of vaccination(s) and infection." As mentioned before, your study was not necessarily designed to look at this effect: Although you did not find a significant reduction, this does not necessarily mean that the sequence and number of immunizing events was not of importance. Please rephrase.

Discussion:

- "The notification of positive tests or vaccination at any time using the study app will have limited recall bias. Moreover, the combination of infection data with serology allowed comparison between patterns of risk of infection and S-antibody concentrations. Still, some misclassification of both infections and vaccinations is possible due to the self-reported nature of these data. Also, the sensitivity for detecting unreported infections by serology will have been less than 100%." The risk of misclassification is more than minimal. I still believe you are overselling the self-reported nature of these data and have to acknowledge this as an important limitation to the data.

Reviewer 2

The authors have responded to my previous comments in a satisfactory manner. There are a few small bits and pieces which would still need to be addressed better.

Abstract:

- "Among participants with hybrid immunity, with one previous pre-Omicron infection, there was no relevant difference in risk of Omicron infection by sequence of vaccination(s) and infection." As mentioned before, your study was not necessarily designed to look at this effect: Although you did not find a significant reduction, this does not necessarily mean that the sequence and number of immunizing events was not of importance. Please rephrase.

Authors: we rephrased this sentence as follows: "Among participants with hybrid immunity, with one previous pre-Omicron infection, we did not observe a relevant difference in risk of Omicron infection by sequence of vaccination(s) and infection."

Discussion:

- "The notification of positive tests or vaccination at any time using the study app will have limited recall bias. Moreover, the combination of infection data with serology allowed comparison between patterns of risk of infection and S-antibody concentrations. Still, some misclassification of both infections and vaccinations is possible due to the self-reported nature of these data. Also, the sensitivity for detecting unreported infections by serology will have been less than 100%." The risk of misclassification is more than minimal. I still believe you are overselling the self-reported nature of these data and have to acknowledge this as an important limitation to the data.

Authors: we rephrased as follows: "Still, misclassification of both infections and vaccinations due to the self-reported nature of these data is possible and therefore a limitation of our study."